# A Novel Design of Water-Activated Variable Stiffness Endoscopic Manipulator with Safe Thermal Insulation

**Qian Gao** [1,2] and **Zhenglong Sun** [1,2,*]

1   School of Science and Engineering, The Chinese University of Hong Kong, Shenzhen, Shenzhen 518000, China; qiangao@cuhk.edu.cn
2   Shenzhen Institute of Artificial Intelligence and Robotics for Society, Shenzhen 518000, China
*   Correspondence: sunzhenglong@cuhk.edu.cn

**Abstract:** In natural orifice transluminal endoscopic surgery (NOTES), an ideal endoscope platform should be flexible and dexterous enough to go through the natural orifices to access the lesion site inside the human body, and meanwhile provide sufficient rigidity to serve as a base for the end-effectors to operate during the surgical tasks. However, the conventional endoscope has limited ability for maintaining high rigidity over the length of the body. This paper presents a novel design of a variable stiffness endoscopic manipulator. By using a new bioplastic named FORMcard, whose stiffness can be thermally adjusted, water at different temperatures is employed to switch the manipulator between rigid mode and flexible mode. A biocompatible microencapsulated phase change material (MEPCM) with latent heat storage properties is adopted as the thermal insulation for better safety. Experiments are conducted to test the concept design, and the validated advantages of our proposed variable stiffness endoscopic manipulator include: shorter mode activation time (25 s), significantly improved stiffness in rigid mode (547.9–926.3 N·cm$^2$) and larger stiffness-adjusting ratio (23.9–25.1 times).

**Keywords:** minimally invasive surgery (MIS); NOTES; surgical robot; variable stiffness endoscope; biocompatible thermoplastic; phase change materials

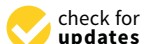

## 1. Introduction

Natural orifice transluminal endoscopic surgery (NOTES), as a paradigmatic renovation of minimally invasive surgery (MIS), is gaining increasing popularity because of its advantages, such as shorter recovery time, avoidance of external trauma and minimized post-surgery discomfort [1,2]. In NOTES, a slender endoscope is required to be flexible to access the stomach or intestine via tortuous natural orifices. Once arriving at the lesion site, the endoscope should be stiff enough to withstand the surgical payloads [3]. Apart from the contradiction between flexibility and rigidity, the dimensions are crucial for endoscopes used for NOTES, as they need to be designed with a suitable outside diameter (OD) to match the narrow human orifices, and a large lumen should be provided to guide or interchange various surgical end-effectors. Particularly, the surgical safety cannot be ignored in endoscope design, due to its direct contact with tissues [4]. However, existing endoscope designs show limited capabilities in achieving trade-offs for these issues, which greatly hinders their extensive clinical applications for NOTES.

Several conceptual designs have been proposed to improve the endoscope towards shifting between flexible and rigid modes. The variable stiffness techniques in surgical manipulators can be categorized into two domains: structure-based and material-based techniques. In the structure-based designs, applied forces are commonly employed to implement stiffness increase by restricting the relative motion between parts. For instance, cable tensions were utilized in [5] to increase the friction between articulated joints in a continuum manipulator, the frictions limited the joint rotations and a rigid endoscope was obtained. An inflated tube arising from the inner water pressure was applied in [6] to

stiffen a spring, and high stiffness was achieved through squeezed threads located between the tube and the spring. The particle blocking principle used in [7,8] could also change the stiffness, and the rigid mode of the endoscope occurred when the internal particles were jammed by the negative air pressure. Although these variable stiffness mechanisms are feasible and can be activated quickly, their bulky bodies make deployment difficult in the usage of NOTES [9].

Thermally activated phase change materials (PCMs), including phase change alloy (PCA) [10–12] and thermoplastics [13–15], have been applied in stiffness adjusting for endoscopes. PCA can be melted to soften the endoscope, and the solidified PCA can switch the endoscope to rigid mode. The alloy of gallium (Ga), indium (In) and tin (Sn) with a high melting point (about 90 °C) was used in [10], and [11] advanced it by using an alloy of bismuth (Bi, 32.5 wt%), indium (In, 51 wt%) and tin (Sn, 16.5 wt%) with a relatively low melting point (62 °C). Yet, these PCAs are not stiff enough to provide desirable stiffness in rigid mode [15]. Another shortcoming in [11] is obvious due to its mode activation using electric heating, as the presence of an electric current (4 A) will cause a surgical safety hazard. The surgical risk also needs to be considered in [12] for the usage of toxic metals (Cerrolow 117) in a variable stiffness catheter. Thermoplastics including polylactic acid (PLA) [13], acrylonitrile butadiene styrene (ABS) [14] and polyethylene terephthalate (PET) [15] were also used for on-demand tuning of stiffness in endoscope designs, and they showed better stiffness-adjusting properties. Once heated above its glass transition temperature, the thermoplastic, that turns to a rubbery state, will possess a low flexural modulus and the flexible mode of the endoscope occurs. On the other hand, the thermoplastic in a glass state can stiffen the endoscope and its rigid mode can be achieved. There still exist constitutive defects of these materials in clinical uses, which can be listed as: (1) The PLA's brittleness in glass state will cause fractures due to the surgical payloads [16]. (2) The ABS has too high glass transition temperature (105 °C) [17]. (3) The PET will produce toxic pyrolysis products if it is heated to a high temperature (70 °C), which is a little over its glass transition temperature (68 °C) [18]. In these designs, they all used electric heating for temperature increases. Except for the hidden surgical risk mentioned above, the electric heating may produce excessive temperature and liquefy the thermoplastics and cause structural damage, which results in challenges for temperature control.

This paper presents a promising variable stiffness endoscopic manipulator for NOTES using a starch-based biocompatible thermoplastic named FORMcard, whose stiffness is significantly tunable with temperature. The selection of FORMcard thermoplastic, among other materials, is largely due to its biocompatibility and relatively low softening temperature (about 65 °C). Hot water (65 °C) and cold water (5 °C) are employed to activate the flexible mode and rigid mode of the endoscopic manipulator, respectively. The thermal insulation made of microencapsulated phase change material (MEPCM), is designed to delay the surface temperature increase in the endoscopic manipulator caused by the hot water supply. The water temperature-dependent mode activation method and the specially designed thermal insulation further improve the safety of our proposed endoscopic manipulator. Finally, the excellent variable stiffness performances of the designed endoscopic manipulator with regard to mode activation time and stiffness in different modes as well as the stiffness-adjusting ratio between the stiffness in rigid mode and flexible mode were validated through experiments.

## 2. Methodology

### 2.1. Materials

Thermoplastic materials are potential candidates that can be employed for stiffness adjustment due to their tunable flexural modulus with temperature [19]. If the temperatures of thermoplastics are raised above their glass transition temperatures, the stiffness-adjusting mechanisms of the softened thermoplastic materials (in a rubbery state) will produce low stiffness. On the contrary, the thermoplastics in a glass state (at temperatures below their individual glass transition temperatures) can contribute to a rigid stiffness-adjusting mechanism.

Table 1 summarizes the properties (including the glass transition temperature and the flexural modulus in a glass state) of commonly used thermoplastics. Among these thermoplastic materials, the styrene-based polymers, such as polystyrene (PS) and ABS, and the chlorine polymers like polyvinyl chloride (PVC) thermoplastic, as well as the fluoropolymers like polytetrafluoroethene (PTFE) thermoplastic, will emit their monomers and pyrolysis products when they are heated to a high temperature (around their glass transition temperatures) [20–23]. These thermal decomposition products, including vinyl chloride ($C_2H_3Cl$), styrene ($C_8H_8$) and tetrafluoroethene ($C_2F_4$), are toxic substances [24,25]. PET thermoplastic (glass transition temperature: 68 °C), in particular, will break down into carcinogenic diethylhexyl phthalate (DEHP), when heated to 70 °C [18,26]. Some other widely used thermoplastics like polycarbonate (PC), polyethylene (PE), polyoxymethylene (POM) and polypropylene (PP) are non-toxic if heated to a high temperature [27,28]. However, they all have too high or too low glass transition temperatures (145–150 °C, −78 °C, −35 °C and 155 °C, respectively) [29–34], which are not suitable for internal use in the human body.

**Table 1.** Properties of commonly used and our applied thermoplastics [20–23,26,29–36].

| Abbreviation | Polymer | Glass Transition Temperature (°C) | Flexural Modulus (GPa) in Glass State |
|---|---|---|---|
| PLA | Polylactic acid | 60–65 | 2.39–4.93 |
| ABS | Acrylonitrile butadiene styrene | 110–125 | 2.07–4.14 |
| PET | Polyethylene terephthalate | 68–80 | 2.41–3.1 |
| PP | Polypropylene | 155 | 0.8–2.17 |
| Nylon 6/6,6 | Polyamide | 47–57/−15–77 | 0.7–2.83 |
| PVC | Polyvinyl chloride | 75–105 | 2.07–3.45 |
| PS | Polystyrene | 80–100 | 3.0–3.6 |
| PTFE | Polytetrafluoroethylene | 130 | 1.14–1.42 |
| PC | Polycarbonate | 145–150 | 2.34 |
| PE | Polyethylene | −78 | 0.84–0.95 |
| POM | Polyoxymethylene | −35 | 2.6–2.88 |
| FORMcard | Starch-based poly | About 65 (measured) | 0.55 (measured) |

Apart from the chemical properties, some thermoplastics' constitutive physical features block their application for endoscopic stiffness adjustment. Taking the PLA thermoplastic as an example, its brittleness shown in the glass state will cause structural damage when withstanding surgical payloads. Nylon plastics, i.e., the polyamide (PA) polymers, have acceptable glass transition temperatures (nylon 6: 47–57 °C, nylon 6,6: −15–77 °C) [35]. The flexural modulus of nylon plastics, however, has a larger range and significantly varies with the humidity [36], which will lead to challenges for future stiffness adjustment. Consequently, the commonly used thermoplastic materials are not suitable for stiffness adjustment in clinically used endoscopes, due to their natural chemical and physical properties.

Peter Marigold, a researcher from London, UK, launched his thermoplastic product named FORMcard on Kickstarter in November 2015. This campaign reached its funding target within the first 24 h and is now sold globally to customers, retailers and distributors all over the world [37]. FORMcard thermoplastics are made of non-toxic bioplastic, which is synthesized from starch-based biodegradable polymers and non-styrene-based pigments [37,38]. In addition, FORMcard products can be reused and were specially designed to fix broken everyday items like toys or fabricate temporary tools [37,39], making use of their thermoplasticity. Thus, the safety of the FORMcard thermoplastic at a high temperature makes it more applicable for the thermal stiffness adjustment of endoscopes, compared with thermoplastics with toxic pyrolysis products.

Figure 1 shows several preliminary property tests for the FORMcard thermoplastic. As displayed in Figure 1a, a force gauge was used to measure the flexural modulus of FORMcard thermoplastic, and the measurement sample is an off-the-shelf FORMcard product, a red card with a thickness of 2.2 mm. The measured flexural modulus of the

FORMcard thermoplastic is 0.55 GPa, which has the same order of magnitude as that of commonly used thermoplastics according to the summary in Table 1.

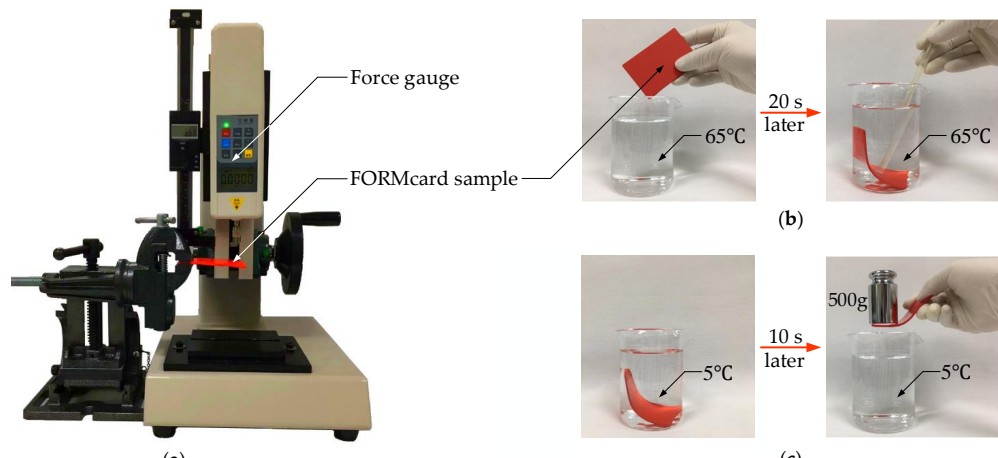

**Figure 1.** Preliminary property tests for FORMcard thermoplastic: (**a**) flexural modulus measurement using a commercial FORMcard product; (**b**) softening test after heating FORMcard thermoplastic sample with water at 65 °C for 20 s; (**c**) stiffening test after cooling FORMcard thermoplastic sample with water at 5 °C for 10 s.

Figure 1b,c display two other tests on the glass transition temperature of the FORM-card thermoplastic. As shown in Figure 1b, we put a FORMcard thermoplastic sample into a beaker containing water at 65 °C. Twenty seconds later, the sample in the beaker was soft like a piece of rubber. Subsequently, the sample was removed and placed in another beaker with water at 5 °C. After being cooled for 10 s, the sample could withstand a 500 g weight (see Figure 1c). Although the glass transition temperature of the FORMcard thermoplastic has not been reported by the designer, we can ascertain that its glass transition temperature threshold should be below 65 °C through the tests exhibited in Figure 1b,c. The sample can be softened without melting at 65 °C, from which the approximate softening temperature of the FORMcard thermoplastic could be confirmed (65 °C) in this study. Compared with the commonly used thermoplastics with non-toxic pyrolysis products listed in Table 1, the FORMcard thermoplastic has stiffness of the same grade as well as a more appropriate softening temperature for internal use in the human body. As a result, the FORMcard thermoplastic is a more applicable candidate material for endoscopic stiffness adjustment and was applied to design a variable stiffness endoscopic manipulator in this research.

*2.2. Structure Design and Working Principle*

A scheme view of the designed variable stiffness endoscopic manipulator is given in Figure 2. The continuum mechanism for stiffness adjustment, as can be seen in Figure 2f, includes a column of disks (made of stainless steel) with equal intervals of 10 mm. These disks are fixed on a nickel–titanium backbone, whose elasticity can help the endoscope return to its original pose (straight shape) after surgical operation.

Figure 2e,f show that the cable-driven manipulator has 2 independent bending segments (proximal and distal segment), which are respectively actuated by 4 cables (4 proximal cables and 4 distal cables). The ends of the cables are respectively fixed at the corresponding pinholes of the disk using knots, with their ends extending out of the manipulator's lumen and connected with the torque output terminal of motors outside the human body. The cables that are uniformly distributed in the circumferential direction of the manipulator and away from the center of corresponding segment can transfer the torques arsing from motors and provide driving moments for endoscopic manipulator. After the endoscopic manipulator is inserted into the human body via natural orifices and arrives at a rough location of the lesion site, the position of a position closer to the

lesion, where the surgical operation can be conducted conveniently. Thus, 4 distal cables are employed for adjusting the pose configuration of the distal segment or the position of the distal disk, but not for providing the axial stiffness of the endoscopic manipulator. The distal cables go through the whole manipulator, and the change in their routes occurs at the transitional segment between the proximal segment and distal segment. As shown in Figure 2e, these 4 distal cables converge close to the center of the proximal segment and scatter around the distal segment, so distal cables will generate driving forces at both the distal segment and proximal segment. However, the pose configuration of the proximal segment embedded inside the orifice of the human body should not be adjusted. Thus, 4 proximal cables away from the center of the proximal segment are used to provide the proximal segment with counterbalanced forces against the forces generated from distal cables, by which the proximal segment will not bend when the distal segment is actuated by the distal cables.

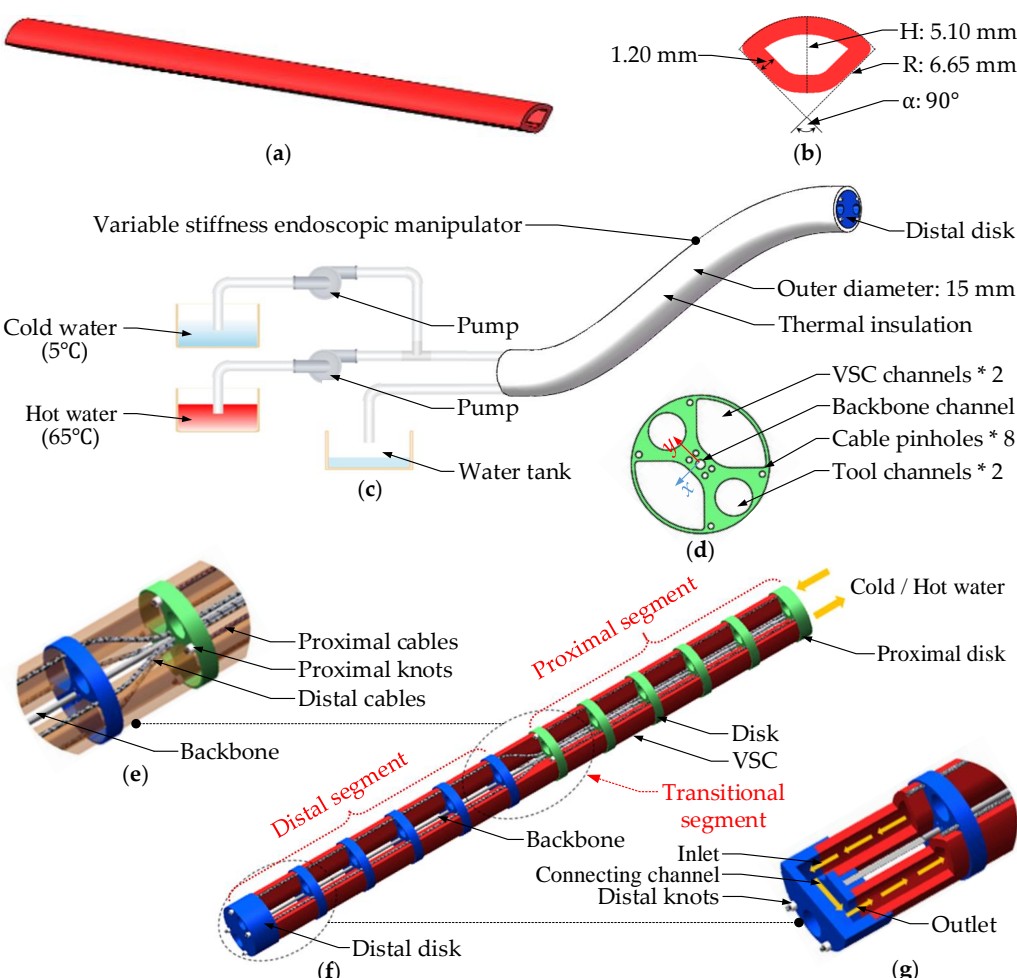

**Figure 2.** Variable stiffness endoscopic manipulator: (**a**) structure of the VSC; (**b**) VSC's fan-shaped cross section; (**c**) working principle and external structure of the endoscopic manipulator; (**d**) disk with thickness of 2 mm and OD of 12.2 mm; (**e**) transitional segment between proximal segment and distal segment; (**f**) continuum stiffness-adjusting mechanism with 2 tool channels of 3.75 mm in diameter; (**g**) connecting channel of these two VSCs.

The stiffness adjustment for the designed endoscopic manipulator mainly relies on the two hollow VSCs made of FORMcard thermoplastic (see Figure 2a), which are located in the VSC channels consisting of several disks and run through the whole endoscopic body. As shown in Figure 2f, the water at different temperatures (5 °C and 65 °C) is injected from the proximal disk into one of the VSCs. After flowing through the VSC and arriving

at the inlet of the connecting channel inside the distal disk, water will be guided along the connecting channel and flow into the lumen of another VSC (see Figure 2g). At room temperature, the VSC is rigid due to the FORMcard thermoplastic being in its glass state, and the current mode of the endoscopic manipulator equipped with VSCs will be rigid mode. If hot water at 65 °C is injected and flows through the lumen of these two VSCs, the heat generated from the hot water will be transferred to the VSCs. Then, the FORMcard thermoplastic, reaching its softening temperature, will convert to its rubbery state, which can soften the VSCs so as to switch the endoscopic manipulator to its flexible mode. Once the cold water (5 °C) is injected, the temperature of the FORMcard thermoplastic in rubbery state will drop sharply and will return to its glass state, and then the VSCs will harden and the rigid mode of the endoscopic manipulator will be restored. It should be noted that a larger cross-sectional area of the VSC can lead to a larger moment of inertia, so as to generate a higher axial bending stiffness of the endoscopic manipulator in rigid mode. Thus, we designed the VSC with a fan-shaped cross-section, whose dimensions are given in Figure 2b. The VSCs are fixed in the holes machined in the disks, and their fit clearance is 0.05 mm. From Figure 2b,d, we can see that these two VSCs can occupy most of the cross-sectional area of the disks, under the condition that these two tool channels (with diameters of 3.75 mm) have enough space to guide or interchange the surgical end-effectors.

To avoid a surface temperature increase in the endoscopic manipulator arising from the injected hot water, thermal insulation is wrapped around the stiffness-adjusting mechanism, as the external structure of the endoscopic manipulator shown in Figure 2c. In addition, the working principle and the operation procedures of the designed variable stiffness endoscopic manipulator are (Figure 2c):

(1) Before surgical operation, the hot water (65 °C) should be constantly pumped into the lumen of the VSCs to soften the endoscopic manipulator (flexible mode activation).

(2) Once the endoscopic manipulator reaches the desired flexible mode, the endoscopic manipulator needs to be inserted into the human body via natural orifices and arrive at the rough lesion position.

(3) Through the driving forces of proximal and distal cables arising from the extracorporeal motors with a preset control program (not shown in this paper), the pose configuration of the distal segment can be adjusted and the distal disk (distal end of the endoscopic manipulator) can be sent closer to the lesion site.

(4) Then, the hot water supply should be terminated, and the cold water (5 °C) is required to be pumped into and flow through the lumen of the VSCs (rigid mode activation).

(5) When the endoscopic manipulator reaches the desired rigid mode, the cold water supply should be stopped.

(6) In the final step, end-effectors should be sent into the lumen of the endoscopic manipulator along the tool channels consisting of multiple sequentially arrayed disks, and they eventually arrive at the distal end of the endoscopic manipulator to carry out surgical operations.

It should be noted that, after the cold or hot water has flowed through the two VSCs, it will eventually flow out of the endoscopic manipulator and be stored in a water tank as shown in Figure 2c. After finishing the above operation procedures, the rigid mode of the endoscopic manipulator is desired to provide sufficient axial stiffness, so that it can play a role as a stable platform for the surgical operation of end-effectors.

*2.3. Thermal Insulation Design*

After flexible mode activation, a constant hot water (65 °C) supply is required to maintain the flexibility of the endoscopic manipulator, so that the surgeon can insert the flexible endoscopic manipulator into the human body. The joule heat arising from the hot water will be transferred to the surface of the endoscopic manipulator and cause the surface temperature to rise. However, the heated surface is in direct contact with the inner wall of the human orifice as the surgeon inserts the endoscope into the body, and excessive

surface temperature will cause burns to human tissues. Therefore, measures must be taken to avoid this surgical risk.

As shown in Figure 2c, the endoscopic manipulator is clad with a thermal insulation. According to the suggestion proposed in [3], the OD of endoscopes in NOTES should not exceed 15.00 mm. Thus, we designed the OD of the thermal insulation to be 15.00 mm, and the thickness of the thermal insulation is 1.40 mm. Figure 3 displays the macroscopic structure of the thermal insulation as well as the microscopic view of its components. As shown in Figure 3a, the thermal insulation can be divided into 3 layers: outer wall, MEPCM layer and inner wall. The outer wall and inner wall are thin rubber films with a thickness of 0.20 mm, and the space between them is filled with MEPCM (filling density: 0.75 g/mL). Macroscopically, the MEPCM shows a powder state composed of solid particles (see Figure 3b,d), and is widely used in the field of biomedical materials because of its good biocompatibility [40]. These solid particles are microcapsules with an OD of 40 μm, consisting of an inner core (60% by volume) and outside shell (40% by volume) in microscopic view (see Figure 3c).

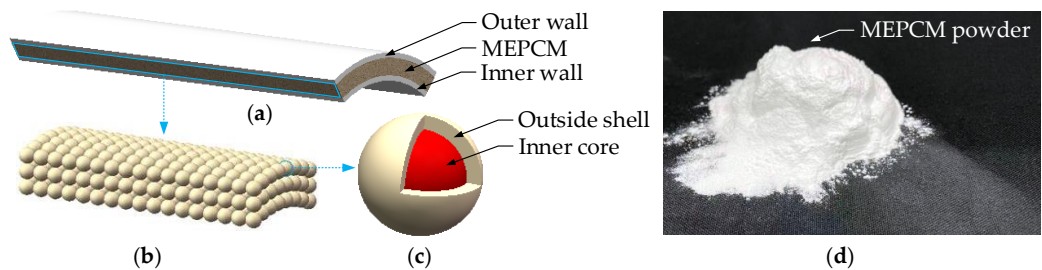

**Figure 3.** Thermal insulation of the variable stiffness endoscopic manipulator: (**a**) layer structure of the thermal insulation; (**b**) macroscopic view of the MEPCM layer with filling desity of 0.75 g/mL; (**c**) microcapsule in microscopic view; (**d**) physical map of the MEPCM powder with stacking density of 0.45 g/mL.

The main chemical composition of the microcapsule inner core is a straight-chain normal alkane ($C_nH_{2n+2}$), i.e., paraffin, and various chain lengths (*n*) of the normal alkane can cause different melting points of paraffin [41]. When heated to its melting point, the paraffin in its original solid state will be liquefied and transformed into the solid–liquid mixed state. With a continuous heat supply, the paraffin in the solid–liquid mixed state will absorb a large amount of heat energy and can maintain its temperature at its melting point for a long time [42]. That is the latent heat storage principle of the MEPCM, making use of the paraffin's solid–liquid phase change properties [43]. The outside shell is generally made of materials with very high melting points and good thermal stability, in order to avoid the flow of the inner core in the solid–liquid mixed state [44].

In this study, the outside shell of the microcapsule is made of crystalline titanium dioxide ($TiO_2$) with a melting point of 1870 °C, and the main component of the inner core is normal alkane ($C_{20}H_{44}$) with a chain length of 20, i.e., paraffin with a melting point of 37 °C (about the temperature of the human body). When the hot water (65 °C) is pumped into the lumen of the endoscopic manipulator and flows through the VSC's lumen, its generated heat energy will be transferred to the thermal insulation. The continuous heat conduction will heat the inner core to its melting point, and the phase state of the inner core will switch to the solid–liquid mixed state. According to the above analysis, the inner core will absorb a large amount of heat arising from the hot water within the lumen of the endoscopic manipulator, and maintain the surface temperature (temperature of the thermal insulation) of the endoscopic manipulator at 37 °C for a long time. Hence, the MEPCM can provide thermal insulation with the capacity of delaying a surface temperature rise, and the surface temperature (37 °C) is an appropriate temperature for the human body. It should be noted that the hot water needs to be constantly provided to maintain the flexibility of the manipulator, before end-effectors are used for the surgical operation. During this process,

therefore, our designed thermal insulation with a latent thermal insulating function can keep the endoscopic manipulator surface at a normal body temperature (37 °C) for a long time, which is able to provide enough safe operation time for a surgeon inserting the flexible endoscopic manipulator into the human body.

## 3. Experiments and Analyses

According to the conceptual design above, we fabricated a prototype of the variable stiffness endoscopic manipulator, whose inner stiffness-adjusting mechanism and the external structure are displayed in Figure 4a,b, respectively. For validating the performances and feasibility of the proposed conceptual design, a mode activation time test, bending stiffness test and surface temperature test were conducted using the fabricated prototype. In order to ensure the replicability of the designed endoscopic manipulator and the reported experimental results, the prototype used for experimental validation was manufactured based on the defined parameters (dimension of each part and specification of each selected material) in Section 2. Furthermore, the experimental results were also analyzed and compared with the existing variable stiffness endoscopic manipulator designs.

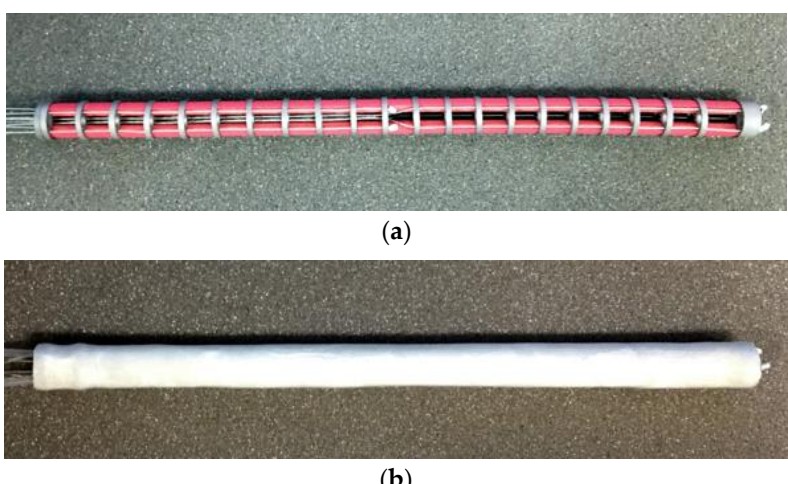

(**a**)

(**b**)

**Figure 4.** Prototype of the designed variable stiffness endoscopic manipulator: (**a**) inner continuum stiffness-adjusting mechanism; (**b**) overview of the fabricated endoscopic manipulator.

### 3.1. Mode Activation Time Test

In order to confirm the activation time of the flexible mode and rigid mode, we switched the prototype of the endoscopic manipulator between these two modes. The time it took for the manipulator to switch from rigid mode to flexible mode was determined as the flexible-= mode activation time, and the required time for the manipulator to switch from flexible mode to rigid mode was considered as the rigid mode activation time. To visually make a distinction between the flexible mode and rigid mode, we applied a load of 150 g at the distal end of the manipulator and observed its shape changes. If a significant shape change occurred due to the applied distal load, the current manipulator was considered to be in its flexible mode. When the applied distal load caused a slight shape change, the manipulator was considered to be in rigid mode. The reason loads were applied at the distal end of manipulator is that the distal end of the manipulator in rigid mode can serve as a stable base for surgical tools to withstand tissue interaction forces during surgical operations, where the surgical operations (such as suture and grab) usually cause lateral surgical payloads. If the endoscopic manipulator is not rigid enough, the lateral surgical payloads will cause shaking of the endoscopic lens, which could be a large disturbance for the surgical operations. Thus, a weight of 150 g was applied at the distal end of the endoscopic manipulator and externally provided the lateral (radial direction of the manipulator) payload for our validation (1.5 N as an average force for the tool–tissue interaction).

In the flexible-mode activation time test (see Figure 5), we conducted flexible mode activation on the straight manipulator that was originally in rigid mode. During the process of flexible mode activation, hot water at 65 °C was constantly pumped (with a flow rate of 180 L/h) from a thermostatic water container into the lumen of the manipulator, and the 150 g weight provided the downward forces at the distal end of manipulator throughout the whole process. With the constant hot water supply, the stiffness of the manipulator would decrease and the manipulator would gradually switch from the initial rigid mode to flexible mode. In addition, due to the presence of the distal load, the shape change of the manipulator would be more and more obvious with the decrease in stiffness of the manipulator.

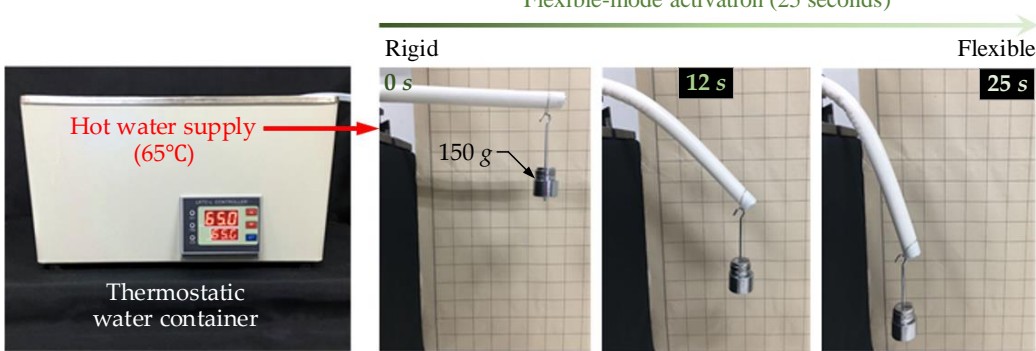

**Figure 5.** Flexible mode activation (from rigid mode to flexible mode) time test with constant hot water (65 °C) supply, shapes of manipulator at 0 s, 12 s and 25 s were recorded.

The shapes of the manipulator at different moments (0 s, 12 s and 25 s) of the flexible mode activation were recorded and are displayed in Figure 5. As can be observed, the manipulator at 0 s (in rigid mode) can hold the applied distal load and showed slight bending. After conducting flexible mode activation for 12 s, the manipulator could be bent to a larger angle (see the second recorded manipulator shape in Figure 5). The final recorded manipulator shape shows that the stiffness of the manipulator dropped dramatically and a significant deformation of the manipulator occurred at 25 s of the conducted flexible mode activation. As a result, the flexible mode activation time of our designed manipulator is defined as 25 s.

Figure 6 displays our implemented rigid mode activation time test, where the manipulator, that was initially in flexible mode, experienced rigid mode activation. Because of the built-in backbone with certain axial elasticity, the manipulator in flexible mode would show a straight shape if without a distal load. If conducting rigid mode activation on a straight manipulator, the stiffness of the manipulator without a distal load would increase and the manipulator would also show a straight shape. In the conducted rigid mode activation time test, just after the manipulator, in its initial flexible mode, experienced rigid mode activation, a load of 150 g was immediately applied at its distal end and the manipulator shape was observed for determination of the desired rigidity. There were three independent experimental steps in the rigid mode activation time test, and they can be described and analyzed as follows:

(1) Figure 6a displays the first experimental step, where cold water (5 °C) was pumped into the lumen of the manipulator that was initially in flexible mode for 5 s. At 5 s, rigid mode activation was terminated and the load of 150 g was immediately applied at the distal end of the manipulator. The recorded manipulator shape shows that the manipulator was significantly buckled by the distal load, which indicates that the manipulator (at 5 s) was not rigid enough and the rigid mode activation time of 5 s was insufficient.

(2) In the second experimental step, as shown in Figure 6b, cold water started to be pumped into the lumen of the manipulator in flexible mode (at 0 s), until 10 s. At 10 s,

the distal load of 150 g was applied and the manipulator showed certain load capacity. However, the presence of the manipulator's large bending angle demonstrates that the time of 10 s is also not long enough to switch the manipulator from flexible mode to the desired rigid mode.

(3) The final experimental step is displayed in Figure 6c, where rigid mode activation was conducted in the manipulator with a duration 20 s. At 20 s, the cold water supply was stopped and the distal load of 150 g was applied at once. As can be observed from the recorded shape of the manipulator with the distal load, the manipulator could withstand the distal load and showed a slight deformation, which indicates that with desired rigidity of the manipulator was obtained.

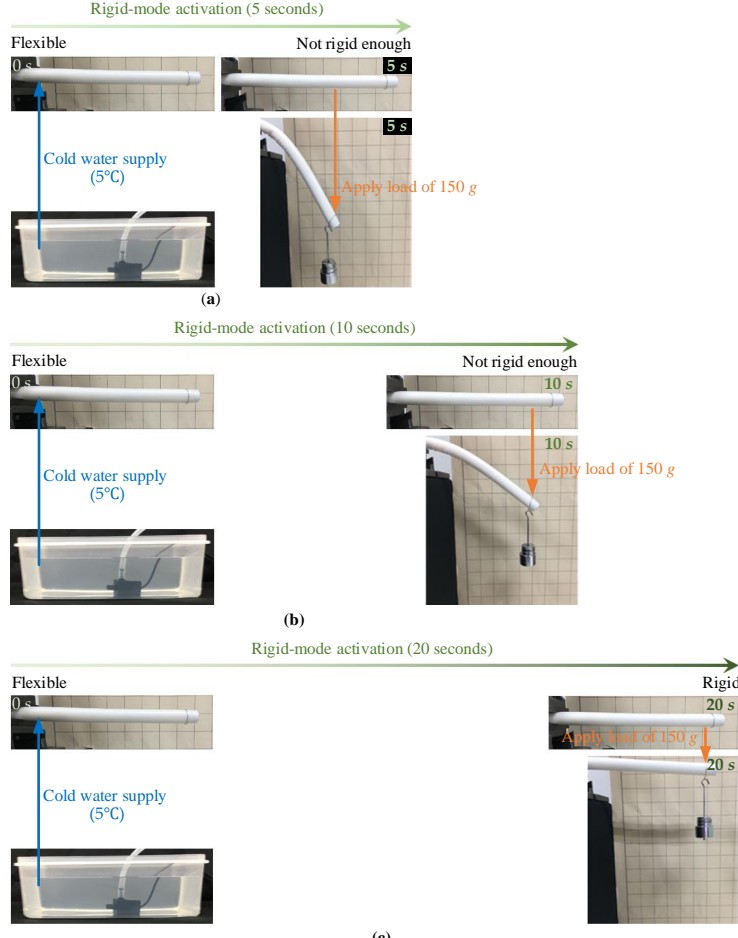

**Figure 6.** Rigid mode activation (from flexible mode to rigid mode) with cold water (5 °C) supply: (**a**) conducting rigid mode activation in manipulator in flexible mode for 5 s, which led to the manipulator not being rigid enough; (**b**) conducting rigid mode activation in manipulator in flexible mode for 10 s, which led to the manipulator not being rigid enough; (**c**) conducting rigid mode activation in manipulator in flexible mode for 20 s, the manipulator showed the desired rigidity.

Through the three conducted experimental steps for the rigid mode activation test, we can observe that if the endoscopic manipulator, originally in flexible mode, experienced rigid mode activation for 20 s, the manipulator was able to reach the desired rigid mode. As a result, the rigid mode activation time of the designed endoscopic manipulator was confirmed (20 s).

Because of the instantly applied external forces, the mode activation speeds of structure-based stiffness-adjusting mechanisms [5–8] are theoretically faster than those of the material-based techniques involving thermal conduction. Thermally stiffness-adjustable endoscopes proposed in [10,11,15] used the mode activation methods of electric heating and ambient

temperature cooling, and mode activation times of 37, 63 and 30 s were reported, respectively, while the longer rigid mode and flexible mode activation time of our designed endoscopic manipulator is only 25 s. By comparison, our design advanced the material-based variable stiffness endoscopic manipulator in terms of the mode activation speed.

After obtaining the flexible mode activation time (25 s measured) as well as the rigid mode activation time (20 s measured), a specification for the application of the designed endoscopic manipulator regarding time response can be made explicitly: (1) before inserting the manipulator into the human body, surgeons should perform the operation of flexible mode activation (hot water supply) for 25 s, so that the endoscopic manipulator in the desired flexible mode can reach the lesion site via the human orifice, and the distal pose configuration can be adjusted on demand. (2) After finishing the adjustment of the distal pose configuration, rigid mode activation (cold water supply) should be conducted with a duration of 20 s, with which the distal end of the manipulator in the desired rigid mode can serve as a stable base for end-effectors. In addition, another specification for actual applications can be reported according to several observations from these conducted tests. As seen from the final recorded manipulator shape in Figure 5, the manipulator in flexible mode could almost reach a bending angle of 90°. Subsequently, the prototype of the endoscopic manipulator could withstand activation between different modes multiple times and bends in the rigid mode activation time test, and structural damage did not occur. Thus, the bending angle range of our designed endoscopic manipulator can be defined as [0, 90°), which is able to meet the criterion of endoscopes for NOTES suggested in [3].

### 3.2. Bending Stiffness Test

For characterizing the stiffness-adjusting performances of the proposed design, bending stiffness tests of the endoscopic manipulator in different modes were carried out. The bending test experiment setup with a simple supported beam structure is shown in Figure 7. In this test, these two supporting spans were set at an interval of 15 cm. A loading pin applied a load to the axial middle position of the manipulator, at a speed of 2 mm/min, until an 8 mm deflection of the manipulator was reached.

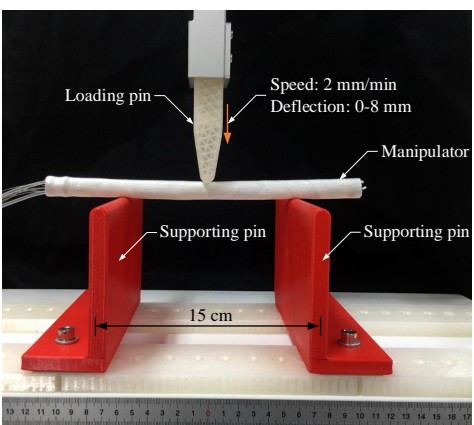

**Figure 7.** Experiment setup for bending stiffness test using the fabricated prototype of the manipulator.

It is worth noting that the designed manipulator has an asymmetrical cross section, and the measured moment of inertia changes with the circumferential load direction of the manipulator. Therefore, we applied bending forces in the circumferential limit directions (the x-axis direction and the y-axis direction as can be seen in Figure 2d), of the manipulator in flexible mode and rigid mode, respectively. A sensor built into the loading pin was employed to record the bending force applied by the loading pin when the deflection of the manipulator occurred, and Figure 8 plots the bending force versus deflection.

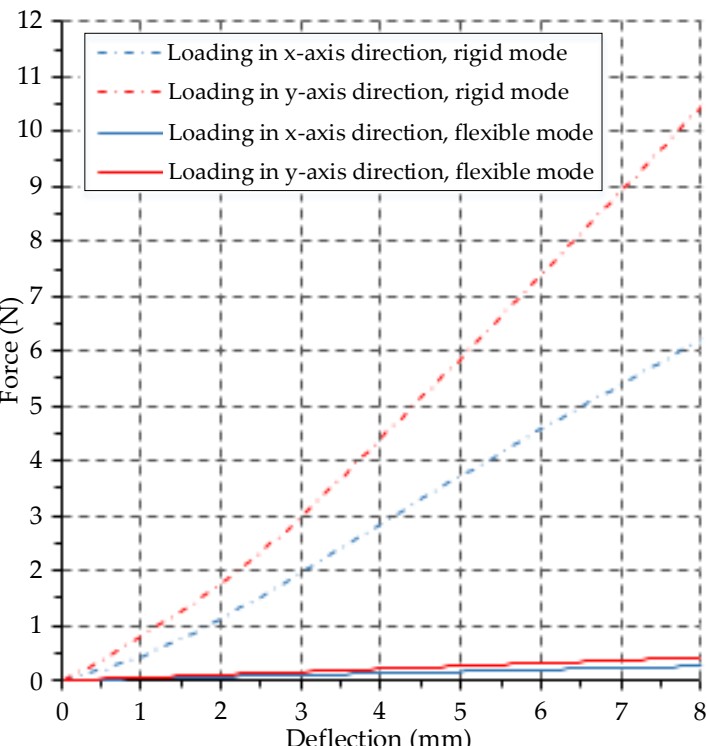

**Figure 8.** Bending stiffness test results for the designed variable stiffness endoscopic manipulator.

According to parameters of the bending stiffness test experiment setup displayed in Figure 7 and the test results shown in Figure 8, the stiffness of the manipulator can be calculated following the simple supported beam bending equation:

$$EI = \frac{FL^3}{48\delta} \tag{1}$$

where $E$, $I$ represent the flexural modulus and the moment of inertia, respectively. In this experiment, $F$ is the bending force recorded by the force sensor, $L$ is the interval of the two supporting pans and $\delta$ is the deflection of the manipulator in the load direction. Let $E_f$, $E_r$, $I_x$ and $I_y$ express the flexural modulus of the manipulator in flexible mode, flexural modulus of the manipulator in rigid mode, moment of inertia on the x-axis and moment of inertia on the y-axis, respectively. For the experimental steps that applied the bending force in the x-axis direction, the calculated stiffness values in flexible mode and rigid mode are $E_f I_x = 22.9\,\mathrm{N\cdot cm^2}$ and $E_r I_x = 547.9\,\mathrm{N\cdot cm^2}$, respectively. With regard to the tests applying the bending force in the y-axis direction, the calculated stiffness values in flexible mode and rigid mode are $E_f I_y = 36.9\,\mathrm{N\cdot cm^2}$ and $E_r I_y = 926.3\,\mathrm{N\cdot cm^2}$, respectively. In addition, the stiffness-adjusting ratio between rigid mode and flexible mode can be obtained using the following expression:

$$\left[\sigma_x, \sigma_y\right] = \left[\frac{E_r I_x}{E_f I_x}, \frac{E_r I_y}{E_f I_y}\right] \tag{2}$$

where $\sigma_x$ and $\sigma_y$ represent the stiffness-adjusting ratio obtained from the stiffness values measured in the x-axis and y-axis directions, and their calculated results are $\sigma_x = 23.9$ times and $\sigma_x = 25.1$ times, respectively. Thus, the stiffness-adjusting ratio of our proposed variable stiffness endoscopic manipulator is 23.9–25.1 times.

Table 2 summarizes the variable stiffness performances of several endoscope designs available in the literature. The structure-based variable stiffness endoscopes proposed in [5,6] theoretically have high stiffness in rigid mode and low stiffness in flexible mode, due to the large external actuation forces and the separated parts caused by the released

actuation forces. A small stiffness-adjusting ratio (1.8 times) of a stiffness-tunable endoscopic manipulator was reported in [8], because their proposed mechanism had relatively higher stiffness in flexible mode caused by the remaining friction between particles. In particular, these structure-based mechanisms have excessive outside dimensions, and their tool channels for end-effectors are designed outside the endoscopic manipulators. Thus, the ODs of their proposed endoscope designs are far beyond the suggested size (OD: 15 mm) in [3] for endoscopes used in NOTES.

**Table 2.** Stiffness-adjusting performances of several endoscopic manipulator designs in the literature [5,6,8,10,11,15].

| Approach | Stiffness-Adjusting Mechanism/Manner | OD (mm) | Built-In Tool Channels | Stiffness in Flexible Mode ($N \cdot cm^2$) | Stiffness in Rigid Mode ($N \cdot cm^2$) | Stiffness-Adjusting Ratio (Times) |
|---|---|---|---|---|---|---|
| Structure based | Cable tension [5] | 18 | No | Low (in principle) | High (in principle) | Large (in principle) |
| | Inflated tube [6] | 15 | No | No data | 1489 $N \cdot cm^2$ | No data |
| | Particle blocking [8] | 30 | No | No data | No data | 1.8 |
| Material based | Alloy tube of Ga, In, Sn [10] | 22 | Yes | 63.8 (calculated) | 262.9 (calculated) | 4.12 (calculated) |
| | Alloy tube of Bi, In, Sn [11] | 13 | No | No data | Lower (in principle) | No data |
| | PET tube [15] | 14 | Yes | 21.33 | 469.33 | 22 |
| Proposed design | VSC made of FORMcard | 15 | Yes | 22.9–36.9 | 547.9–926.3 | 23.9–25.1 |

A material-based tunable stiffness platform for single-site surgery was proposed in [10], and stiffness tests were conducted with experiment setups with a cantilever beam structure. Their reported test results were: the variable stiffness mechanism with cantilever length of 15 cm and distal load of 5 N showed deflection of 88.2 mm in flexible mode and deflection of 21.4 mm in rigid mode. According to their reported test results and the cantilever beam bending equation ($EI = FL^3/3\delta$), the calculated stiffness values of their designed variable stiffness mechanism in flexible and rigid mode were only 63.8 $N \cdot cm^2$ and 262.9 $N \cdot cm^2$, respectively, and the obtained stiffness-adjusting ratio following Expression (2) was just 4.12 times. The designed endoscopic manipulator for NOTES in [11] had a smaller OD (13 mm), and significant deformation occurred when the manipulator in rigid mode received a 1000 gr ($\approx$64.8 g) load with a 12 cm cantilever length. Our proposed manipulator with a cantilever length of 22 cm could hold a load of 150 g and showed slight bending (see the recorded shape at 0 s in Figure 5), which indicates the higher stiffness in rigid mode of our designed endoscopic manipulator. In the state-of-the-art literature [15], a variable stiffness endoscopic manipulator especially designed for NOTES was proposed. This endoscopic manipulator was slightly more compliant in flexible mode than ours. However, our design had a larger stiffness-adjusting ratio, and showed a great improvement in the stiffness of rigid mode.

Consequently, our proposed endoscopic manipulator has better stiffness-adjusting performances, compared with existing endoscope designs. In addition, the stiffness of a medical endoscope for NOTES is generally in the range of 160–240 $N \cdot cm^2$ [45]. The stiffness of our given design in flexible mode and rigid mode is different from this suggested range, which indicates the great variable stiffness properties of our proposed endoscopic manipulator.

### 3.3. Surface Temperature Test

Apart from the mode activation time test as well as the bending stiffness test, a surface temperature test was conducted for the sake of surgical safety and instructions for the actual application of our proposed variable stiffness endoscopic manipulator. When a surgeon inserts the endoscopic manipulator (in flexible mode) into the human body and

adjusts the distal pose of the flexible manipulator, the manipulator surface is in direct contact with the inner wall of the human orifice. However, hot water (65 °C) is required to be continuously supplied to maintain the flexibility of the manipulator, and a temperature increase on the manipulator surface will happen due to the thermal conduction. The surface temperature of the manipulator in flexible mode is important in surgical safety, since an excessive surface temperature may cause burns to the inner tissues. Thus, the surface temperature changes with a hot water (65 °C) supply were tested, to determine the length of time it took for the surface temperature to increase to a burn temperature threshold. After measuring the time length, we could provide instructions regarding the safe operation time, within which surgeons should finish inserting the manipulator into the human body and distal pose configuration operations.

As shown in Figure 9a, a thermal sensor attached to the manipulator surface was applied to record the surface temperature change with hot water supply. At an ambient temperature of 25 °C, hot water at 65 °C was consistently pumped into the lumen of the manipulator via one of the hot water pipes, with a flow rate of 180 L/h. Simultaneously, the hot water flowed through the two VSCs and flowed out of the lumen of the manipulator via another hot water pipe.

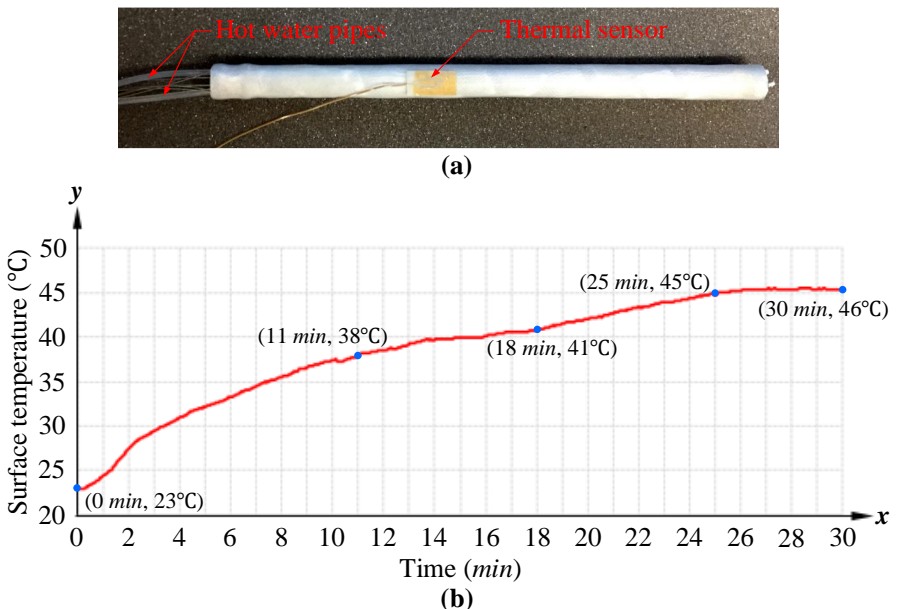

**Figure 9.** Surface temperature test for our proposed variable stiffness endoscopic manipulator: (**a**) established experimental setup, where the fabricated prototype of the manipulator had a thermal sensor for recording surface temperature and was connected with two pipes for hot water (65 °C) supply and discharge; (**b**) endoscopic manipulator surface temperature change recorded from beginning of heat supply to reaching thermal equilibrium.

From the beginning of hot water supply to the state of constant surface temperature (thermal equilibrium), the manipulator's surface temperature was recorded by the mounted thermal sensor, and Figure 9b plots the recorded endoscopic manipulator surface temperature versus time. According to the surface temperature change within 30 min, four observations were made:

(1) When the hot water was pumped in, the surface temperature rose rapidly from 23 °C to 38 °C, in the first 11 min.
(2) As the temperature reached 38 °C, the temperature increase (from 38 °C to 41 °C) was delayed for a long time (from 11 min to 18 min).
(3) Then, slight and fast temperature increase occurred again due to the constant hot water supply, and the surface temperature increased by 4 °C from 18 min to 25 min.

(4)    Finally, the heat conduction reached a balance and the surface temperature was stable at about 46 °C.

Based on the principles of the proposed mode activation approach and the designed thermal insulation, four subsequent observations corresponding to the four observations above were made:

(1)    In the initial 11 min, the heat conduction could be divided into two parts. Firstly, the heat arising from the hot water was transferred from the inner wall of the VSCs to their surface, which was the process of flexible mode activation. Then, the heat was transferred from the VSC surface to the thermal insulation.

(2)    With the temperature increase of the thermal insulation, the microcapsule inner core (paraffin) of the MEPCM reached its melting point and turned into the solid–liquid mixed state, so the surface temperature increase was delayed due to the latent heat storage property of MEPCM.

(3)    From 18 min to 25 min, because of the continuous heat supply, the inner core in the original solid–liquid mixed state was liquefied completely, and the MEPCM lost its heat storage function, so another stage of temperature increase occurred in this process.

(4)    Finally, the heat from the supplied hot water continued to be transferred through the VSCs, air within the manipulator lumen and the thermal insulation to the outside of the manipulator, and this thermal conduction system eventually reached its heat equilibrium.

Similar surface temperature tests (at almost the same ambient temperature) for variable stiffness endoscopic manipulators using thermal activation methods were conducted in [11,15], and they all showed relatively sharp surface temperature increases within a short period of time. With the PTFE coating material used in [11], it took only 36.7 s for the surface temperature to rise from 25 °C to 43 °C and with the aerogel insulation coating applied in [15], it took twice as long, about 70 s, for the surface temperature to increase from 27 °C to 33 °C. For our designed thermal insulation approach, it took almost 19.5 min (from 1.5 min to 21 min) for the surface temperature to rise from 25 °C to 43 °C, and 3 min (from 2 min to 5 min) to rise from 27 °C to 33 °C. It is obvious that our proposed thermal insulation approach shows much slower heat conduction, resulting in a slower surface temperature increase. Although there were slight differences in the test boundary conditions, the better thermal insulation capability of our approach could be verified.

In this experiment, the surface temperature was maintained below 41 °C for about 18 min, while the temperature threshold for tissue burns is 44 °C [46], so the safe surface temperature of our designed endoscopic manipulator can last at least 18 min. Even when taking the mode activation time (25 s measured above) into account, the safe operation time is still about 17 min 35 s. The obtained safe operation time can be regarded as a guide for the surgeon to maneuver the manipulator (in flexible mode) into the human body and locate the lesion site.

## 4. Conclusions and Future Work

In this study, we developed a complete conceptual design of a variable stiffness endo­scopic manipulator with safe thermal insulation for NOTES, making use of the thermomechan­ical properties of FORMcard thermoplastic and the latent heat storage property of MEPCM. Compared with the widely used thermoplastics, the starch-based FORMcard thermoplastic with biocompatibility at a high temperature and relatively low softening temperature is safer for thermal stiffness adjustment in endoscopic manipulators. The novel water temperature-dependent mode activation method can eliminate the clinical risks that exist in the widely used approaches involving electric heating. The specially designed thermal insulation proved that it can provide a long safe operation time for surgeons, which has not been verified in other related literature where thermally stiffness-adjustable endoscopic manipulators were

described. These design concepts based on surgical safety highlight its superiority over existing material-based variable stiffness endoscopic manipulators.

The conducted validation tests show that our variable stiffness endoscopic manipulator has a shorter mode activation time, higher stiffness in rigid mode and larger stiffness-adjusting ratio, compared with the existing endoscopic manipulators designed for NOTES. The excellent stiffness-adjusting properties demonstrate the high potentialities of the given conceptual design for future medical uses. In the next step, we will develop a fully functional endoscope equipped with embedded cameras and other required electronics, and then test it in realistic surgical scenarios (including ex vivo or in vivo endoscopic procedures) before wider clinical applications.

**Author Contributions:** Conceptualization, Q.G. and Z.S.; methodology, Q.G.; software, Q.G.; validation, Q.G.; formal analysis, Z.S.; investigation, Q.G.; resources, Q.G.; data curation, Q.G.; writing—original draft preparation, Q.G. and Z.S.; writing—review and editing, Q.G. and Z.S.; visualization, Q.G.; supervision, Z.S.; project administration, Z.S.; funding acquisition, Z.S. All authors have read and agreed to the published version of the manuscript.

**Funding:** The authors would like to acknowledge the support from the Shenzhen Science and Technology Program (Grant No. RCYX20200714114736115).

**Institutional Review Board Statement:** Not applicable.

**Informed Consent Statement:** Informed consent was obtained from all subjects involved in the study.

**Data Availability Statement:** Please contact Qian Gao (qiangao@cuhk.edu.cn).

**Acknowledgments:** The authors acknowledge all the reviewers for their valuable comments as well as the editing of this article conducted by MDPI services.

**Conflicts of Interest:** The authors declare no conflict of interest.

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
