# Peer review of "A Novel Design of Water-Activated Variable Stiffness Endoscopic Manipulator with Safe Thermal Insulation"

_actuators, doi:10.3390/act10060130_

Round 1

Reviewer 1 Report

This work presents an interesting water-activated actuator, which is expected to be used as an endoscopic manipulator. The paper presents the application of Natural Orifice Transluminal Endoscopic Surgery (NOTES) and introduces concepts of Thermally activated Phase-Change Materials and Thermoplastic materials. In this sense, the FORMcard material is introduced, which is the base of the proposed actuator. The paper has presented the design of an endoscopic manipulator, which includes a thermal insulation design. Afterward, some experiments are presented to characterize the performance of the device. Those experiments demonstrate results about surface temperature, times of activation, and bending stiffness. The article is well writing. However, I have some concerns about the experiments performed and actual application of this actuator, as follows:

  1. The authors should mention the specifications required for these applications regarding time response and kinematics. It is unclear the range of motion needed for this application, and the authors should provide an experimental section to measure the range of motion. This is very relevant to warranty the controllability of this actuator.
  2. In this sense, the bending presented with a load of 150 grams represents the range of motion somehow. The authors should give information in an actual application on how they will provide these forces externally to modify the position of the end effector.
  3. The authors should present information on how it is expected to be controlled the motion of the actuator and verified experimentally. Otherwise, it is hard to understand the future applicability of this device.
  4. Figure 5 does not make sense at all. I recommend putting in only one XY-axis figure put all the range of the function. They should put the axis from 0 to 50 minutes.
  5. Following the temperature experiment, the stabilized temperature is over 40 degrees, which is very dangerous for most human tissues. The authors should adequately reflect on the applicability of this surface temperature, which notably reduces the application of this work in real scenarios.
  6. The temperature experiment also should represent the hysteresis of the system embodying the behavior during warming and cooling stages.
  7. Something that is also unclear is the replicability of this actuator. If it is not defined correctly, they should replicate all the experiments with at least three different actuators.
  8. In the experiments, it is mentioned that the experiments were replicated three times. The author should represent all the measurements with a media value and a standard deviation.
  9. The authors should reduce the lines of the paragraphs, and the document has extensive sections that make the text difficult to follow. There are many main ideas in the same paragraphs.
  10. The citations are defined at the beginning of the sentence. It confuses the reading (e.g., Lines 38 and 40).
  11. Figure 1 corresponds to experiments provided in this article, so it should be moved to the methodology section.

Reviewer 2 Report

In this paper, a novel design of variable stiffness endoscopic manipulator whose stiffness can be thermally adjusted by using a new bioplastic named FORMcard is proposed. The water at different temperatures is employed to activate the manipulator between rigid mode and flexible mode. The conducted validation tests show that the variable stiffness endoscopic manipulator has shorter mode-activation time, higher stiffness in rigid mode and larger stiffness-adjusting ratio, compared with the existing endoscopic manipulators designed for NOTES. On the whole, the demonstrations are detailed, accurate and professional, the authors’ work is interesting and valuable, and the paper is acceptable.

Author Response

Thanks for your proposed evaluations, we believe that further improvements are going to be achieved in our future work.

Reviewer 3 Report

The article presents a current topic. It is well done and therefore I recommend it for publication. Thanks to the authors for the great work. 

Author Response

Thanks for the given comments, we are going to further improve our existing work with your encouragements.

Round 2

Reviewer 1 Report

I agree with the clarifications of the cover letter and the paper. I recommend the paper for publication.